# DDosTC: A Transformer-Based Network Attack Detection Hybrid Mechanism in SDN

**DOI:** 10.3390/s21155047

**Published:** 2021-07-26

**Authors:** Haomin Wang, Wei Li

**Affiliations:** School of Control and Computer Engineering, North China Electric Power University, No. 2 Beinong Road, Changping District, Beijing 102206, China; liwei@ncepu.edu.cn

**Keywords:** software-defined networking, transformer, convolutional neural network, DDoS, hybrid model

## Abstract

Software-defined networking (SDN) has emerged in recent years as a form of Internet architecture. Its scalability, dynamics, and programmability simplify the traditional Internet structure. This architecture realizes centralized management by separating the control plane and the data-forwarding plane of the network. However, due to this feature, SDN is more vulnerable to attacks than traditional networks and can cause the entire network to collapse. DDoS attacks, also known as distributed denial-of-service attacks, are the most aggressive of all attacks. These attacks generate many packets (or requests) and ultimately overwhelm the target system, causing it to crash. In this article, we designed a hybrid neural network DDosTC structure, combining efficient and scalable transformers and a convolutional neural network (CNN) to detect distributed denial-of-service (DDoS) attacks on SDN, tested on the latest dataset, CICDDoS2019. For better verification, several experiments were conducted by dividing the dataset and comparisons were made with the latest deep learning detection algorithm applied in the field of DDoS intrusion detection. The experimental results show that the average AUC of DDosTC is 2.52% higher than the current optimal model and that DDosTC is more successful than the current optimal model in terms of average accuracy, average recall, and F1 score.

## 1. Introduction

With the complexity of network architecture and the rapid growth of the connection requirements of Internet-connected devices, the traditional complex Internet architecture (complex control, complex software, high line costs, and difficult expansion) cannot dynamically handle modern network applications. Therefore, modern network applications require a scalable architecture which should be able to provide reliable and adequate services based on specific traffic types [1]. Software-defined networking (SDN) [2] is a new network structure that has emerged in recent years; its scalability, dynamics, and programmability can simplify the complex traditional Internet architecture and its real-time characteristics can meet high-availability requirements [3]. This kind of network structure separates the control plane of the network from the data-forwarding plane, both to realize programmable control of the underlying hardware through the software platform in the centralized controller, and to realize the flexible deployment of network resources on demand. However, this centralized management method [4] causes the SDN controller to become a single compromise point for the entire network. Compared with the traditional network structure, it is more vulnerable to attacks that cause the entire network to collapse. A denial-of-service (DoS) [5] attack is a malicious attempt in which the attacker generates a large number of data packets or requests, eventually overwhelming the target system. Distributed denial-of-service (DDoS) [6] is a special form of a DoS-based denial-of-service attack. Multiple attackers in different locations simultaneously launch attacks on one or several targets, or one attacker controls multiple machines at different locations and uses these machines to attack victims simultaneously in a distributed and coordinated large-scale attack [7]. However, it is difficult to detect DDoS attacks using bot devices, so the detection of DDoS attacks by intrusion-detection systems has become a challenging task.

Recently, SDN has been widely used in various Internet of Things systems, and in the realization of the Internet of Things, new-generation communication (5G) plays an important role. The realization of 5G also needs SDN, which is a currently a developing technology [8]. The reliability of its underlying network directly affects the overall reliability of the system. The Internet of Things is very easily attacked via DDoS [9]. Further, the nature of SDN increases the number of DDoS attacks [8]. Therefore, ensuring the security of SDN is of great significance to the security of these systems, which is one of the main issues often discussed in relation to the Internet of Things technologies.

In recent years, transformers [10] based on the attention mechanism have achieved great success in text classification, dialogue tasks, and other natural language-processing (NLP) fields. The main method is to pre-train on a large text corpus and then fine-tune on a smaller task-specific dataset [11]. Due to the computational efficiency and scalability of transformers, they have not only been used in the field of natural language processing but also have a worthy role in the direction of image classification in the field of computer vision [12].

Inspired by a transformer’s text classification and computer vision image classification in NLP, He et al. and Bikmukhamedo et al. applied a transformer to traffic classification [13,14] and achieved good results.

This paper proposes a deep learning technology based on a transformer and a CNN DDoS attack-detection model (DDosTC). We combine the CNN and the transformer to apply DDoS attack detection. In addition, the control plane plays an important role in SDN; it is the brain of the network, the south network control, and the north business support. Consequently, this paper has implemented their detection module at the SDN control plane. The architecture diagram and DDoS attack-detection module of SDN are shown in Figure 1. Compared with the current optimal model, the model proposed herein shows better performance in terms of accuracy, recall, F1 score, and AUC indicators.

The main contributions of this work are as follows:We propose a transformer-based deep learning framework (DDosTC) that uses the computational efficiency and scalability of a transformer, and the predictive ability of a convolutional neural network, to detect distributed denial-of-service attacks on SDN;In experiments, we use the newly released dataset, CICDDoS2019, which contains a variety of DDoS attacks and bridges the gap in the existing current dataset, to evaluate our model;We test several state-of-the-art deep learning frameworks in detecting DDoS attacks. We evaluate our proposed model in terms of accuracy, recall rate, F1 score, and AUC. The method we propose has the best performance;We further evaluate the performance of the model by randomly dividing the training set and the test set.

The structure of the rest of the paper is as follows: Section 2 introduces related work; Section 3 introduces our proposed technology; Section 4 introduces the new dataset and the evaluation of our proposed model; and finally, Section 5 summarizes the paper and proposes future work.

## 2. Related Work

Machine learning has been widely used in intrusion detection in SDN environments in recent years. This section introduces the latest model methods used to detect DDoS attacks in the SDN environment.

Early in [15], NN + biological danger theory was used to mitigate DDoS attacks in SCN. The authors of [16] and [17] used RBF-SVM (a combination of nonlinear classification (RBF) and linear classification (SVM)) for DDoS detection; the accuracy rates in these studies reached 95.11% and 97.60%, respectively. The authors of [18] used multiple linear SVM + SOM; the accuracy rate here reached 97.6% with a false positive rate of 3.85%. It can be seen that the detection effect of machine learning is more successful than traditional methods.

As the performance of deep learning in many fields has far surpassed that of traditional ML methods, and deep learning does not require feature engineering, good performance can be achieved by directly inputting data into the network for analysis and processing. Therefore, in the SDN environment, deep learning is already widely used in DDoS attack-detection tasks. The authors of [19] proposed a DDoSNet framework that applied RNN to the detection of DDoS attacks and achieved 99% accuracy with an AUC of 98.8%. In addition, [20] used the deep CNN ensemble framework to detect DDoS attacks and achieved 98% accuracy in the ISCX2012 dataset [21]. The use of a hybrid deep learning model for intrusion detection was explored in [22], which used a hybrid model of LSTM, CNN, and GRU applied to the CICIDS2017 dataset [23]; the highest accuracy obtained was 98.97% [24]. Using the SAE deep learning method to extract features for DDoS detection, the 8-class and 2-class accuracy rates were 95.65% and 99.82%. This was also used in [4] on the CICDDoS2019 [25] dataset to propose an SDN defense system against distributed denial-of-service attacks and intrusion attacks, using GRU deep learning methods to perform multidimensional (multistream feature) analysis to detect attacks on the SDN controller. The average result in terms of accuracy, precision, recall rate, and F-measure rate was 99.94%. The authors of [26] used GRU-RNN to test the NSL-KDD [27] dataset. The authors of [28] proposed a deep learning-supported intrusion detection and prevention system (DL-IDPS) using a deep learning model to achieve an accuracy of 98.3% against DDoS attacks.

Due to their rapid development, [13] transformers were applied to the classification of encrypted traffic and [14] also conducted experiments on network traffic classification. The performance of the classifier was on average 4% higher than that of the integrated one.

Through the above review, we found that most of the studies in the literature used earlier datasets to carry out experiments. In order to effectively realize the detection of the latest types of attacks, in this article we used the CICDDoS2019 dataset, which contains the latest various types of DDoS attacks. We found that there is still room for improvement in the detection performance reported in the above-mentioned documents, so we included the popular transformer framework to propose a new detection model to achieve effective detection of DDoS attacks.

## 3. Model Description

This section introduces our proposed DDosTC model and elaborates on the detection process of this model against DDoS attacks.

### 3.1. Convolutional Neural Network

A convolutional neural network (CNN) [29] is a feed-forward neural network in which artificial neurons can respond to surrounding units, and it has achieved good results in image classification and face recognition. Convolutional neural networks mainly include three network structures:A convolutional layer to extract features;A pooling layer, used for feature dimensionality reduction;A fully connected layer, mainly used for classification.

The data features are extracted through the convolutional layer, and then the dimensionality is reduced through the features of the pooling layer. The number of data and parameters are compressed, over-fitting is reduced, and the fault tolerance of the model is improved. The features obtained through the previous convolution and pooling layers are classified in the final fully connected layer.

### 3.2. Transformer

The transformer network architecture was proposed by Ashish Vaswani et al. in the article “Attention Is All You Need” [8]. It achieved good results in NLP applications. The encoder and decoder of this network architecture did not use network architectures such as RNN or CNN. They used an architecture that completely relied on the attention mechanism [30] and the network also introduced a multi-head attention mechanism.

A transformer is mainly composed of an encoder and decoder. Each encoder block has two layers: multi-head attention and feed-forward. A decoder has three layers: multi-head attention, multi-head attention (encoder-decoder attention), and feed-forward.

The encoder input was changed into three parts: query, key, and value through input embedding. First, positional encoding was used to address the fact that an attention mechanism itself cannot capture position information. The transformer’s unique multi-head attention mapped the query and key to high dimensions. The similarity was calculated in different subspaces $\left(\beta_{1},\beta_{2},\ldots ,\beta_{n}\right)$ of space β so that the expressiveness of each layer of attention could be enhanced without changing the number of parameters, and the output of the feed-forward layer was passed to the decoder.

The decoder conversion result, corresponding to the encoder, was also converted into three parts: query, key, and value as input. After positional encoding, masked multi-head attention, feed-forward, and the output structure from the encoder were jointly input to multi-head attention and then passed through feed-forward, which first output the space, and then finally output the probability distribution through the dense layer. For each block, the transformer added the normalization layer, which contained the residual structure and layer normalization to standardize the optimization space and accelerate convergence.

### 3.3. The Transformer-Based Network Attack Detection Hybrid Mechanism

In this paper, we combine the transformer and CNN to form a new hybrid model, DDosTC, which can achieve better performance than traditional deep learning algorithm models and traditional hybrid models. A structural diagram of the DDosTC model is shown in Figure 2.

We divide the DDosTC model into three parts as a whole: the transformer Layer, CNN layer, and dense layer. A flow chart is shown in Figure 3.

#### 3.3.1. Transformer Layer

In the transformer layer of the DDosTC model, we retain the original structure of the transformer, and the encoder includes multi-head attention, feed-forward, and add and norm. The decoder includes multi-head attention, multi-head attention (encoder-decoder attention), add and norm, and feed-forward. However, the input data are different from those used in the field of natural language processing, so we set the input and output in the encoder and decoder to the same input, with the preprocessed data used as the input. After embedding, the input is divided into three parts—query, key, and value—and the attention mechanism is supplemented by positional encoding because it cannot capture the position information by itself. Then, it is output to multi-head attention; to improve performance, we set all multi-head attention to two heads. Then, in the encoder module, the output of the two-head attention layer and the feed-forward network are controlled by the output of the two-head attention layer and the feed-forward network. This is sent to two add and norm layers for processing to ensure the stability of the data and feature distribution, and to accelerate the optimization speed of the model.

For the decoder module, after the multi-head attention and add and norm, the output result of the encoder is combined and input into a layer of multi-head attention, through space transformation of FFN after two layers of add and norm optimization, and finally output through the dense layer. The activation functions used in the transformer layer are all ReLU functions, defined as
(1)f(x)=max(0,x)

#### 3.3.2. CNN Layer

In the CNN layer, we retain the convolutional layer and the pooling layer; we enter the output results from the transformer layer into the convolutional layer for data feature extraction, then reduce the dimensionality of the pooling layer features, and finally obtain the features. For the fully connected layer, we replace it with global-average pooling (GAP) [31] because GAP does not require a large number of training or tuning parameters, reduces spatial parameters, has a better anti-overfitting effect, and converts between the feature map and the final classification. It is simpler and more natural.

#### 3.3.3. Dense Layer

We established a total of three dense layers (Units = 64, 32, 1), and passed the output results of the CNN layer through two dense layers. The activation functions were all ReLU. In the last dense layer, the activation function used by our model was the Sigmoid function, defined as
(2)$$S\left(x\right)=\frac{1}{1+e^{-x}}$$

## 4. Results and Evaluation

This section introduces the results of experiments with DDosTC and compares our proposed method’s performance with that of the detection methods that have appeared in recent years—recurrent neural network (RNN), gated recurrent unit (GRU), convolutional neural network (CNN), long short-term memory (LSTM), hybrid deep learning (LSTM + GRU), and bidirectional long short-term memory (bidirectional GRU)—in terms of accuracy, recall rate, F1 score, AUC, and other aspects of our proposed model.

In this paper, we used the CICDDoS2019 dataset, which contained the latest common DDoS attacks and was a real dataset that was similar to real-life attacks. There were different modern reflective DDoS attacks in this dataset including PortMap, NetBIOS, LDAP, MSSQL, UDP, UDP-Lag, SYN, NTP, DNS, and SNMP. This dataset contained data for two days. The first day was 12 January 2019, which was used as the training day; this dataset contained 12 different DDoS attacks. The second day was used as the test day; this dataset contained six different types of DDoS attacks.

### 4.1. Evaluation Metrics

We used various indicators to evaluate our proposed model, such as accuracy, recall rate, precision, and F1 score, to conduct systematic benchmark analysis against other related methods. These indicators are commonly used in intrusion-detection systems and are defined as follows:(3)$$\mathrm{Precision}=\frac{TP}{TP+FP}$$
(4)$$\mathrm{Accuracy}=\frac{TP+TN}{TP+TN+FP+FN}$$
(5)$$\mathrm{Recall}=\frac{TP}{TP+FN}$$
(6)$$F1\_\mathrm{score}=\frac{2\times \;\mathrm{Precision}\;\times \;\mathrm{Recall}\;}{\;\mathrm{Precision}\;+\;\mathrm{Recall}\;}$$
where true positive (*TP*) and true negative (*TN*) represent the values that are correctly predicted. In contrast, false positives (*FP*) and false negatives (*FN*) indicate misclassified events.

To test our model’s ability to recognize samples at a certain threshold, we used the ROC curve (receiver operator characteristic curve) [32] and evaluated it by calculating the AUC (area under the ROC curve). The larger the AUC the better, as this value suggested that the diagnostic value of the model was higher.
(1)AUC ≈ 1.0: The most ideal inspection index;(2)AUC 0.7–0.9: High test accuracy;(3)AUC 0.5: The test had no diagnostic value.
(7)$$AUC=\frac{\sum_{k\;\mathrm{posttiveClass}}\;{\mathrm{rank}}_{i}-\frac{M\left(1+M\right)}{2}}{M\times N}$$
where *M* is the number of positive samples and *N* is the number of negative samples.

### 4.2. Experimental Environment

Our experimental operations were conducted on a Linux 64-bit operating system, an Intel Xeon E5-2680 processor, and an NVIDIA GeForce RTX 3080, with 64GB of memory. The experimental environment language was Python, the deep learning framework was TensorFlow, and the main packages used were Numpy, Pandas, Tensorflow, Keras, Sklearn, Pydot, and Matplotlib.

The used BATCH_SIZE was 64, EPOCHS was 20, and the LEARNING RATE was 0.1.

### 4.3. Data Preprocessing

The CICDDoS2019 dataset we used contained 87 extracted IP flow feature data, as shown in Figure 4. Before training, we needed to preprocess the data:We first removed the features of the source and destination IP, source and destination port, timestamp, and flow identification, because we only needed to train the model through packet characteristics. We also removed infinite data and (NaN) data. The final input feature number was 76, and then the data of the remaining features were formatted;We coded the tags, coding all DDoS attack tags to 1 and normal traffic tags to 0.

### 4.4. Experimental Tests and Results

In reality, attack scenarios and attack types change daily. To show the performance of our proposed framework, we divided the dataset into a training set and test set as follows:(1)Training set/test set = 8:2 (randomly selected 80% of the data on the training day and 20% of the data on the test day);(2)Training set/test set = 7:3 (randomly selected 70% of the data on the training day and 30% of the data on the test day);(3)Training set/test set = 6:4 (randomly selected 60% of the data on the training day and 40% of the data on the test day).

We analyzed the accuracy, precision, F1 score, and recall, and the results are shown in Figure 5. When we reduced the training set/test set ratio, we found that, in the caes of the accuracy, F1 score, and recall, their performances decreased, while the performance of precision increased. We found that the more data in the training set, the more effective the training. It can be observed that when the ratio of the training set to the test set was 8:2, the highest accuracy was 0.9982. It is not difficult to see that the average performance was the most successful when the training set/test set ratio was 7:3. In addition, we changed the number of iterations of the model, the number of epochs, the hidden layer activation function, and the number of hidden layers, and found that when the epoch number was 20, the performance was most successful when the hidden layers (except the CNN layer and the transformer layer) contained three layers. When the number of epochs was 20, as the number of hidden layers increased, the amount of model training increased but the accuracy did not change significantly. In addition, we changed the learning rate of the model and found that the best performance was obtained when the learning rate was 0.1.

To verify the superior performance of our model, we compared it with transformer (T), transformer + LSTM + CNN (TLC), and transformer + LSTM (TL) for different training set/test ratios. Details are shown in Table 1.

From Figure 6, Figure 7 and Figure 8, we can see that as the complexity of the model continued to increase, all evaluation indicators dropped significantly. When we only used the transformer model, the performance was inferior to that of DDosTC and TL. For the two hybrid models DDosTC and TL, we also found that the average performance of our model remained slightly higher than that of the TL model. In this sense, the performance of our model was more convincing.

Next, we analyzed the accuracy, precision, F1 score, and recall rate of our model and other models (Table 2). Figure 9, Figure 10 and Figure 11 show the results for different training/testing set ratios and comparisons of the accuracy, precision, F1 score, and recall rate with other models on the same dataset.

From Figure 8, Figure 9 and Figure 10 and Table 1, we can see that the accuracy of DDosTC with a training set/test set ratio of 8:2 was 0.9982; with a training set/test set ratio of 7:3 was 0.9978; and with a training set/test set ratio of 6:4 was 0.9970, which was the most successful when compared with the other models.

The precision for a training set/test set ratio of 8:2 was 0.9988; for a training set/test set ratio of 7:3 was 0.9989; and for a training set/test set ratio of 6:4 was 0.9998. It was evident that this performance was superior to that of the other models.

The F1 score for a training set/test set ratio of 8:2 was 0.9992; for a training set/test set ratio of 7:3 was 0.9989; and for a training set/test set ratio of 6:4 was 0.9984, which were more successful values than those achieved by other models.

In terms of recall rate, the DDosTC model was not optimal when the train/test ratio was 6:4; B-GRU, LSTM, and LSTM + GRU achieved 0.9983, exceeding DDosTC. However, for train/test ratios of 8:2 and 7:3, it was obvious that DDosTC far exceeded the other models.

In general, our model, DDosTC, was superior to the other tested algorithm models. To further verify that our model was more successful than other models, we tested the AUC of the model; we still divided the dataset into three parts, and the results are shown in Figure 12, Figure 13 and Figure 14.

When the training set/test set ratio was 8:2, DDosTC’s AUC score was 99.95%; among the other models, LSTM was greatest at a value of 97.94%, so our model’s score was 2.01% greater.

When the training set/test set ratio was 7:3, the AUC score of DDosTC was 99.86%; among the other models, GRU was greatest at a value of 97.99%, so our model’s score was 1.87% greater.

When the training set/test set ratio was 6:4, the AUC score of DDosTC was 99.90%; among the other models, the greatest value was 97.95% for LSTM, so our model’s score was 1.95% greater.

From all of our experimental results, we can see that, compared with the two models LSTM and GRU, which had the best average performance for intrusion detection, based on the AUC, our model’s average performance was 2.52% and 2.85% greater, respectively.

## 5. Conclusions and Future Work

In the SDN environment, due to the very aggressive nature of DDoS attacks and the constant changes in these attacks, a highly accurate and adaptable intrusion-detection model is required. In this article, we proposed a transformer-based DDosTC model based on the latest DDoS attack dataset, CICDDoS2019, and we simulated the constantly changing randomness of this type of attack. This model and the latest existing DDoS attack-detection model achieved the highest performance in accuracy, precision, and F1 score. Although our model was not optimal in terms of recall rate when the train/test ratio was 6:4, after a comprehensive evaluation, our model’s performance was far superior to that of other models, and our model’s AUC scores were greater, showing its high evaluation ability. In future work, we will try to apply the model to multilabel classification and our model to other datasets for testing.

## 6. Patents

This section is not mandatory but may be added if there are patents resulting from the work reported in this manuscript.

## Figures and Tables

**Figure 1 sensors-21-05047-f001:**
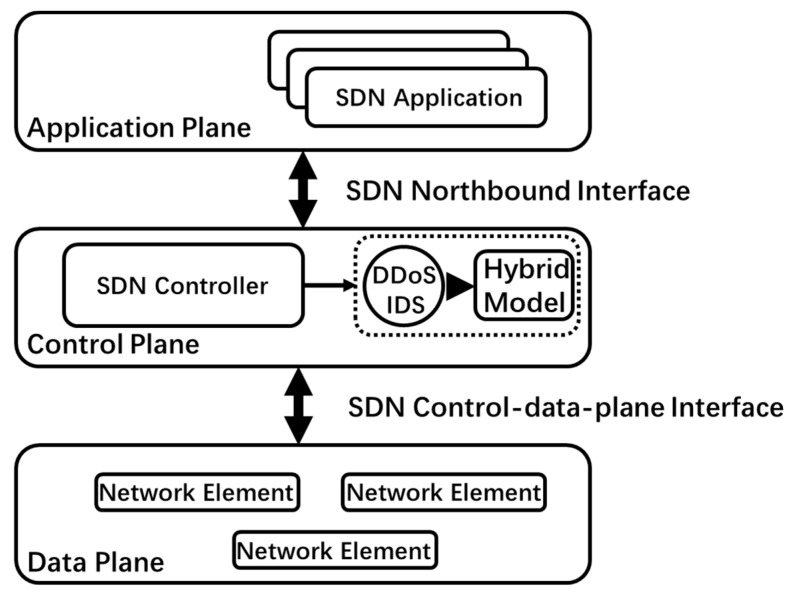
SDN architecture based on DDoS attack-detection module.

**Figure 2 sensors-21-05047-f002:**
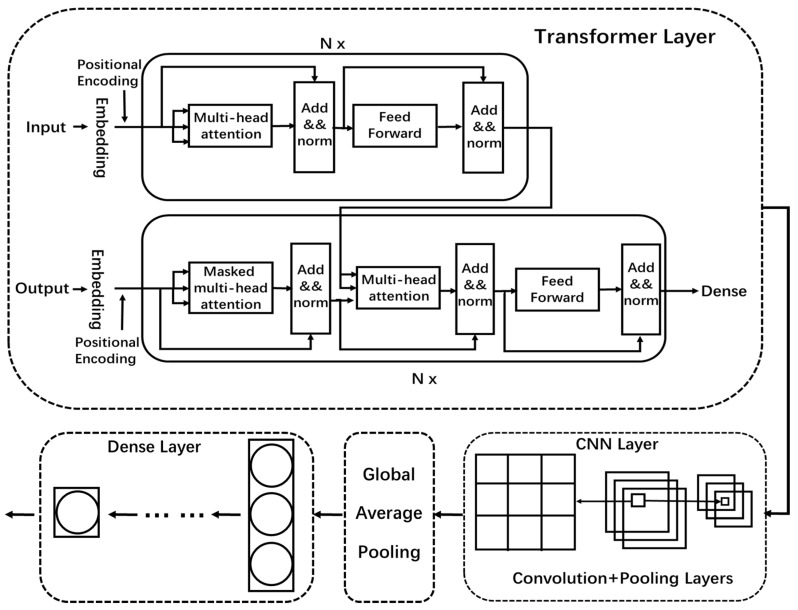
DDosTC overall architecture.

**Figure 3 sensors-21-05047-f003:**
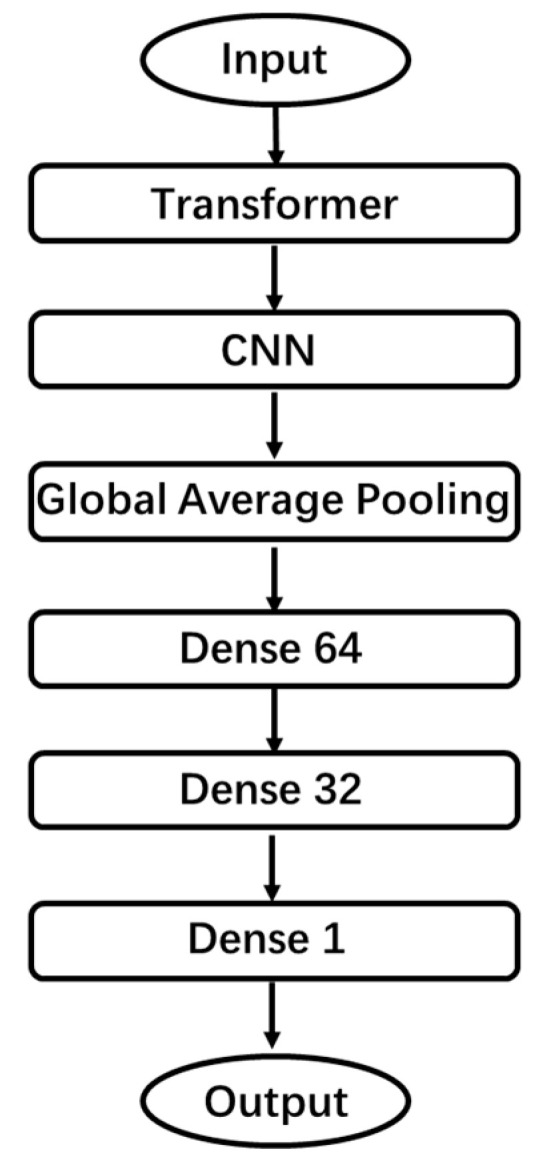
DDosTC flow chart.

**Figure 4 sensors-21-05047-f004:**
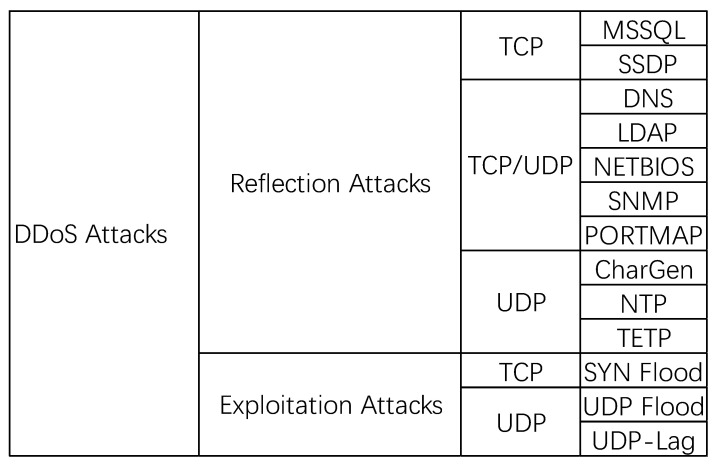
Detailed classification of DDoS attacks in CICDDoS2019.

**Figure 5 sensors-21-05047-f005:**
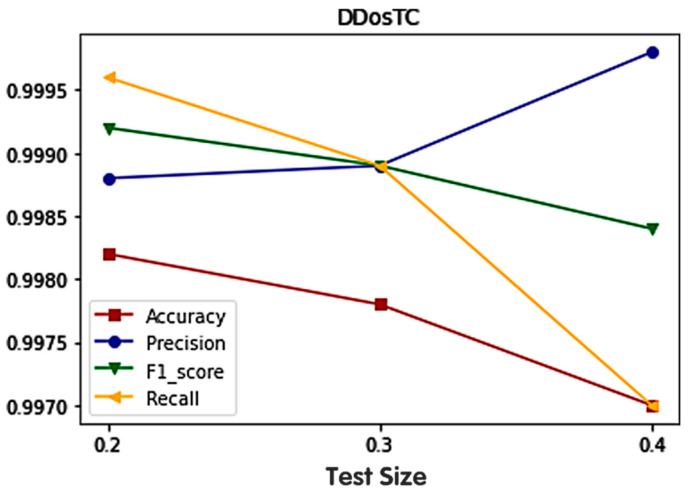
The results of different divisions of the the training set and test set in terms of accuracy, precision, recall rate, and F1 score.

**Figure 6 sensors-21-05047-f006:**
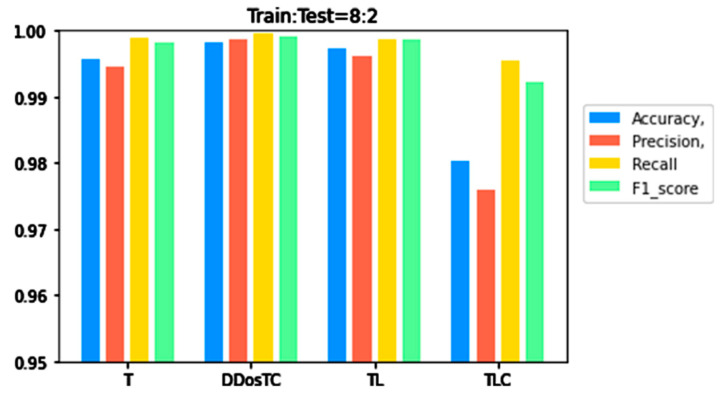
A performance comparison of DDosTC and transformer, tranformer + CNN + LSTM, and transformer + LSTM (Train/Test = 8:2).

**Figure 7 sensors-21-05047-f007:**
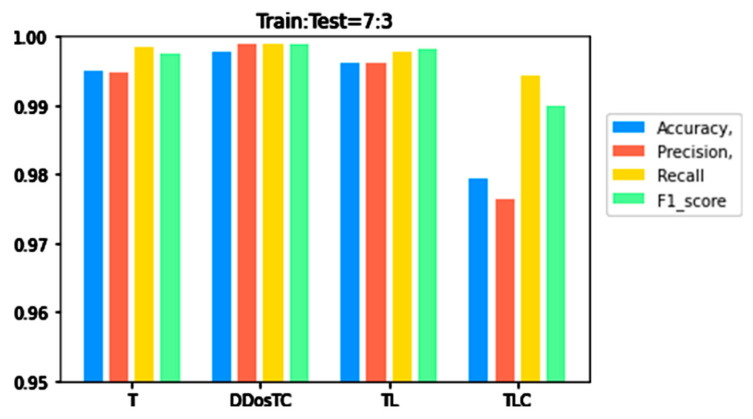
A performance comparison of DDosTC and transformer, transformer + CNN + LSTM, and transformer + LSTM (Train/Test = 7:3).

**Figure 8 sensors-21-05047-f008:**
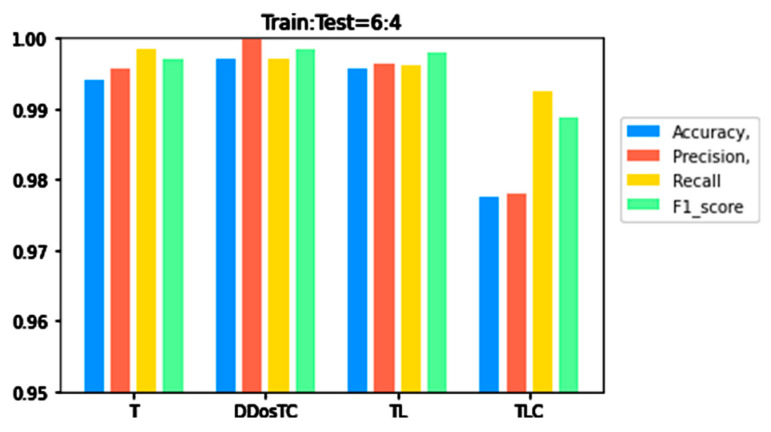
A performance comparison of DDosTC and transformer, transformer + CNN + LSTM, and transformer + LSTM (Train/Test = 6:4).

**Figure 9 sensors-21-05047-f009:**
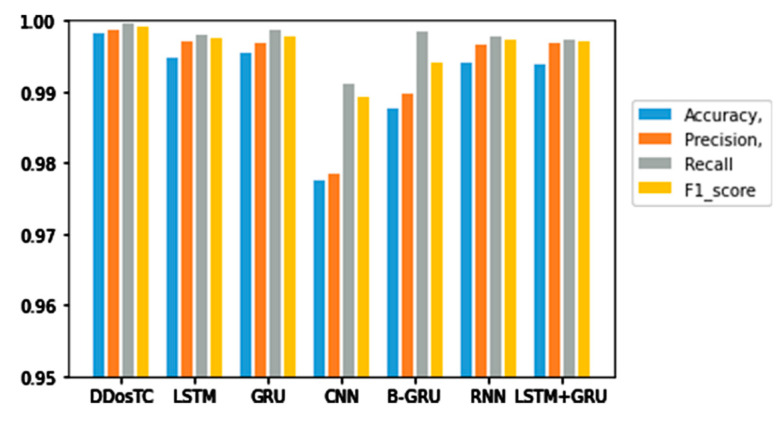
Accuracy, precision, recall rate, and F1 score results for DDosTC and other test methods with a training set/test set ratio of 8:2.

**Figure 10 sensors-21-05047-f010:**
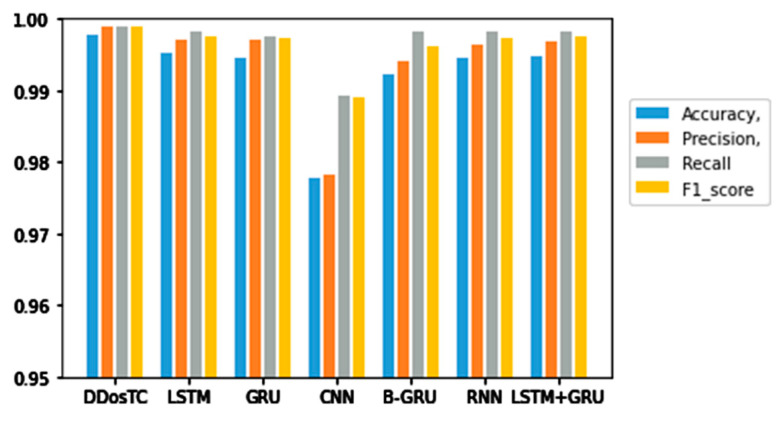
Accuracy, precision, recall rate, and F1 score results for DDosTC and other test methods with a training set/test set ratio of 7:3.

**Figure 11 sensors-21-05047-f011:**
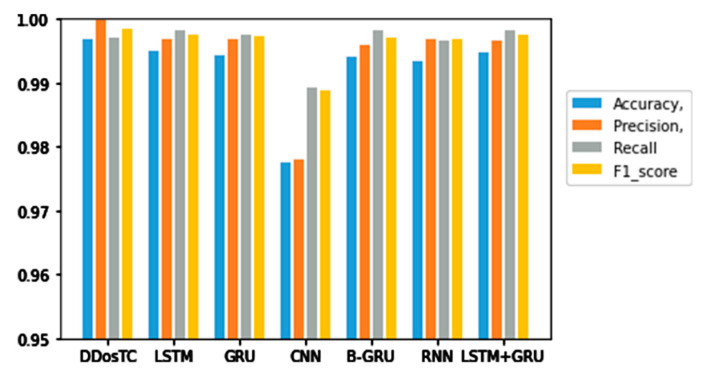
Accuracy, precision, recall rate, and F1 score results for DDosTC and other test methods with a training set/test set ratio of 6:4.

**Figure 12 sensors-21-05047-f012:**
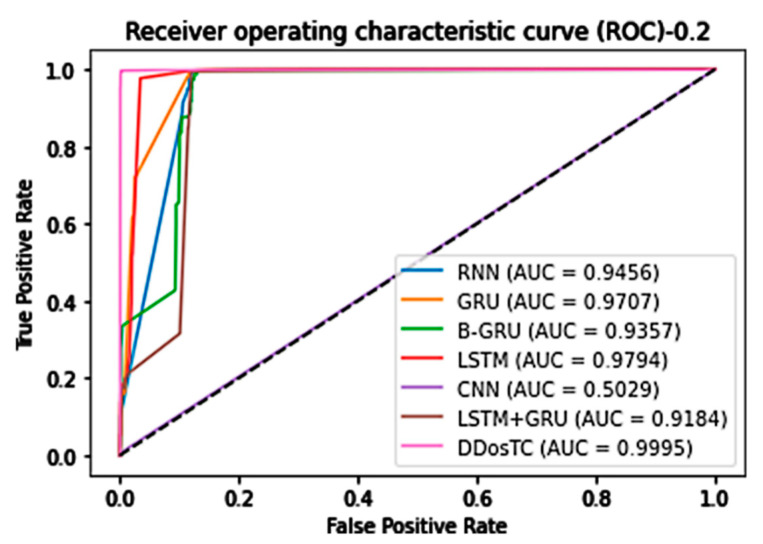
AUC for train/test = 8:2.

**Figure 13 sensors-21-05047-f013:**
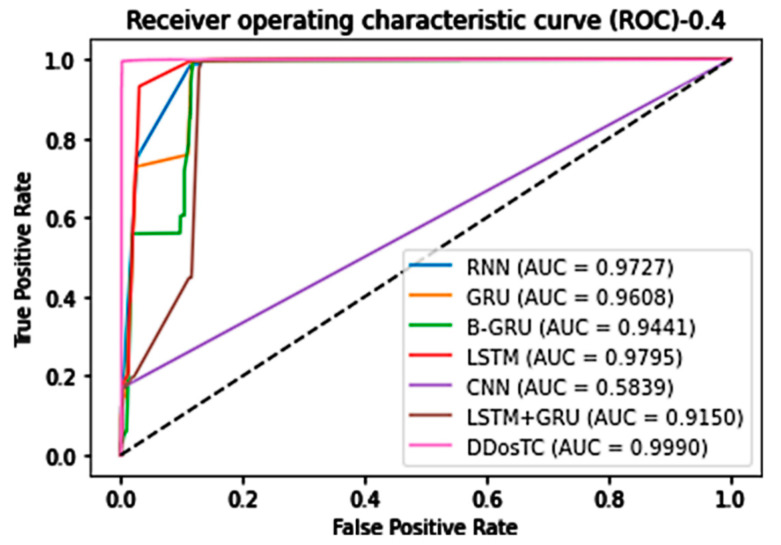
AUC for train/test = 7:3.

**Figure 14 sensors-21-05047-f014:**
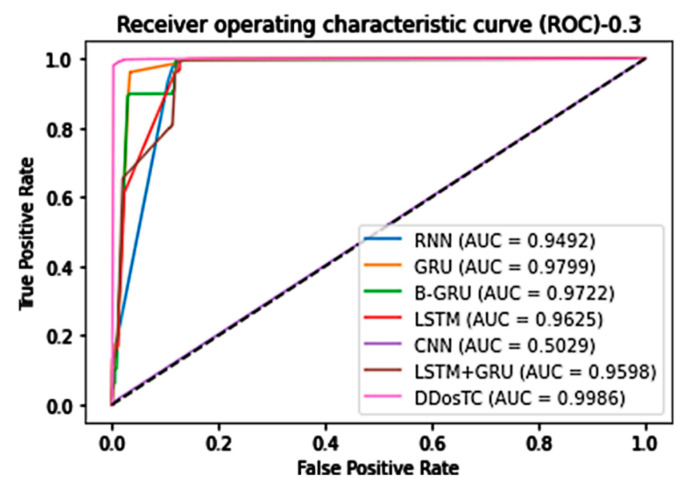
AUC for train/test = 6:4.

**Table 1 sensors-21-05047-t001:** Detailed technical description of hybrid algorithms based on transformers.

Hybrid Algorithm	Test Size	Epochs	Learning Rate	Batch Size
DDosTC	0.2	20	0.1	64
	0.3			
	0.4			
Transformer	0.2	20	0.1	64
	0.3			
	0.4			
Transformer + LSTM	0.2	20	0.1	64
	0.3			
	0.4			
Transformer + CNN + LSTM	0.2	20	0.1	64
	0.3			
	0.4			

**Table 2 sensors-21-05047-t002:** A detailed comparison of DDosTC and other models. The bold numbers are our work.

Data Set Division	Algorithms	Accuracy	Precision	F1 Score	Recall
Train/Test = 8:2	DDosTC	**0.9982**	**0.9988**	**0.9992**	**0.9996**
	GRU	0.9955	0.9970	0.9979	0.9987
	CNN	0.9775	0.9786	0.9892	0.9912
	B-GRU	0.9876	0.9897	0.9941	0.9984
	RNN	0.9942	0.9967	0.9972	0.9978
	LSTM + GRU	0.9939	0.9968	0.9971	0.9974
	LSTM	0.9949	0.9972	0.9976	0.9981
Train/Test = 7:3	DDosTC	**0.9978**	**0.9989**	**0.9989**	**0.9989**
	GRU	0.9945	0.9971	0.9973	0.9975
	CNN	0.9777	0.9782	0.9890	0.9892
	B-GRU	0.9922	0.9942	0.9962	0.9983
	RNN	0.9946	0.9965	0.9974	0.9983
	LSTM + GRU	0.9948	0.9968	0.9975	0.9982
	LSTM	0.9952	0.9970	0.9976	0.9983
Train/Test = 6:4	DDosTC	**0.9970**	**0.9998**	**0.9984**	0.9970
	GRU	0.9944	0.9969	0.9972	0.9976
	CNN	0.9775	0.9781	0.9889	0.9892
	B-GRU	0.9940	0.9959	0.9971	**0.9983**
	RNN	0.9935	0.9969	0.9968	0.9966
	LSTM + GRU	0.9948	0.9967	0.9975	**0.9983**
	LSTM	0.9950	0.9969	0.9976	**0.9983**

## Data Availability

The data repository is available at https://www.unb.ca/cic/datasets/ddos-2019.html. Data was accessed on 5 March 2021.

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
