# Peer review of "DDosTC: A Transformer-Based Network Attack Detection Hybrid Mechanism in SDN"

_sensors, 2021, doi:10.3390/s21155047_

Round 1
Reviewer 1 Report
Paper discusses relevant topic of SDN security and proposes a novel concept to detect DDos attacks, relevant for SDNs in variety of systems, including Internet of Things, where security issues are frequently discussed in the last decade.
Some improvements shall be made to the manuscript to increase its clarity and readability. I hope my suggestions will be helpful and contribute to increase of manuscript overall quality.
I think the introduction explains the context well, just there are some issues to fix.
“For actual experiments, we use the newly released data set CICDDoS2019” … provide a link or citation to this dataset.
“The structure of the rest of the thesis is as follows” … thesis, really? I think it shall be “of the paper”
Change “A denial of service (Dos)” to “A Denial of Service (DoS)”, as well, “Software-defined networking (SDN)” to “Software-defined Networking (SDN)”
In text, instead of “[11,12] applied Transformer to traffic classification” write “Author1 et al. and Author 2 et al. applied Transformer to traffic classification [11,12]”.
Structure of the paper: Instead of “second part” use “Section 2” .. etc.
A suggestion to comment on the relevance of the paper for the research community: Recently, SDNs are used in various types of IoT systems, where the reliability of the underlying network impacts the overall reliability of the system. Hence securing SDNs has implications to security of these systems, which is frequently discussed as one of major problems of IoT technology.
Some literature to support the discussion:
Mahmoud, R., Yousuf, T., Aloul, F., & Zualkernan, I. (2015, December). Internet of things (IoT) security: Current status, challenges and prospective measures. In 2015 10th International Conference for Internet Technology and Secured Transactions (ICITST) (pp. 336-341). IEEE.
Bures M., Klima M., Rechtberger V., Ahmed B.S., Hindy H., Bellekens X. (2021) Review of Specific Features and Challenges in the Current Internet of Things Systems Impacting Their Security and Reliability. In: Trends and Applications in Information Systems and Technologies. WorldCIST 2021. Advances in Intelligent Systems and Computing, vol 1367. Springer.
Related work – explain RBF-SVM
3.1. CNN … don’t use abbreviations in the section headings. The same applies for “3.3. DDosTC” and following
The Transformer network architecture was proposed by Ashish Vaswani and others 137
in the article "Attention Is All You Need". … a citation is essential here
Figure 1 – Transformation layer details cannot be read in the figure and the figure itself is blury
Relu functions,the formula is (1): … rephrase to “Relu functions, defined as:”
Figure 2 – Dense layers have to be distinguished.
Generally, more detailed description is needed to give the reader opportunity to fully understand ad reproduce the describe method.
Experimental results:
Figure 4 – the graph shall be rescalled to point out the differences in more readable way
Rename “4.5. Experimental evaluation” to “results discussion”, or something more fitting
The experimental section shall be finished by a compact summary of the result and conclusion, in which aspects the presented method outperforms the compared alternative. This can be done in Conclusion section, which is rather brief – a summary of the results supported by the major data from the experiments shall be given here.
Threats to Validity section must be added to the paper. In this section, discuss possible limits and sources of bias in the performed experiments and the countermeasures taken to minimize such a possible influence.
The paper shall be proofread by a native speaker to improve the English level. Also, there is number of minor mistakes and typos in the manuscript, some of them I listed in my review. Thorough proofread is needed.
Reviewer 2 Report
It is evident that authors did not read the paper after preparing the text since there are "default" title of the Table 1, strange structure of sentences, many strangely Caps words in the text, bad text formatting and syntax errors on pictures.
There is no relations between SDN technology and the research because the dataset CICDDos2019 does not have any SDN specifics.
The quality of some figures is very low.
The main question to authors is about the need of enhancement the detection quality from 98% to 99%, as can be seen in Table 2. Is this really important result if the only object of it's application is a laboratory dataset?
The problem of DDoS attack detection was developed well enough and improvement of the classification quality looks overabundant.
Other publications display the quality at level of 99% on datasets. It is obvious that nobody needs to enhance such results more.
The term SDN is used without real understanding of the technology it means.
Conclusion is meaningful enough: "our model has AUC score compared with other models." This means that the proposed model has no significant advantages over others.
Round 2
Reviewer 1 Report
Authors have reflected the majority of my comments satisfactorily.
Still, there are some issues to fix:
Figure 1 is blurry and captions inside the schemas merely impossible to read.
The same applies to Figure 4.
And the following graphs. I think it is some formatting problem in the PDF.
In the literature, some references are incomplete, for instance in the last one (Review of Specific Features and Challenges in the Current Internet of Things Systems… ), conference is missing, it shall be In: Trends and Applications in Information Systems and Technologies. WorldCIST 2021. Advances in Intelligent Systems and Computing, vol 1367. Springer., as mentioned in the original review. I have found it for some other references as well, for instance Bikmukhamedo, R. F. , and A. F. Nadeev . "Generative transformer framework for network traffic generation and classification." (2020).
Reviewer 2 Report
1) Very poor quality of all figures
2) The relation between chosen dataset CICDDoS2019 and attacks to SDN is not proven. I do not see any reason to mention SDN in the paper. It's just a DDoS detector. No SDN-specific features were used in the paper.
3) The need of almost 99.999...9% quality of detection for the exact task is not argumented anyhow. All ML methods which give >90% of quality on model dataset are good enough to be used in practice. What is the reason to use this exact approach?
Round 3
Reviewer 2 Report
They improved the quality of figures.
The answers to the second and the third questions are acceptable.